# Immune and spermatogenesis-related loci are involved in the development of extreme patterns of male infertility

Miriam Cerván-Martín[1,2,28], Frank Tüttelmann [3,28], Alexandra M. Lopes[4,5,6,28], Lara Bossini-Castillo[1,2], Rocío Rivera-Egea[7,8], Nicolás Garrido[8,9], Saturnino Lujan[9], Gema Romeu[9], Samuel Santos-Ribeiro[10,11], José A. Castilla [2,12,13], M. Carmen Gonzalvo[2,12], Ana Clavero[2,12], Vicente Maldonado[14], F. Javier Vicente[2,15], Sara González-Muñoz[1,2], Andrea Guzmán-Jiménez[1,2], Miguel Burgos[1], Rafael Jiménez[1], Alberto Pacheco[8,16], Cristina González[8], Susana Gómez[8], David Amorós[8], Jesus Aguilar[8], Fernando Quintana [8], Carlos Calhaz-Jorge[17], Ana Aguiar [17], Joaquim Nunes[17], Sandra Sousa[17], Isabel Pereira [17], Maria Graça Pinto[18], Sónia Correia[18], Josvany Sánchez-Curbelo[19], Olga López-Rodrigo[19], Javier Martín[20], Iris Pereira-Caetano[21], Patricia I. Marques[4,5], Filipa Carvalho[4,22], Alberto Barros[4,22], Jörg Gromoll[23], Lluís Bassas[19], Susana Seixas[4,5], João Gonçalves[21,24], Sara Larriba[25], Sabine Kliesch[26], Rogelio J. Palomino-Morales[2,27,29 ✉] & F. David Carmona [1,2,29 ✉]

We conducted a genome-wide association study in a large population of infertile men due to unexplained spermatogenic failure (SPGF). More than seven million genetic variants were analysed in 1,274 SPGF cases and 1,951 unaffected controls from two independent European cohorts. Two genomic regions were associated with the most severe histological pattern of SPGF, defined by Sertoli cell-only (SCO) phenotype, namely the MHC class II gene *HLA-DRB1* (rs1136759, P = 1.32E-08, OR = 1.80) and an upstream *locus* of *VRK1* (rs115054029, P = 4.24E-08, OR = 3.14), which encodes a protein kinase involved in the regulation of spermatogenesis. The SCO-associated rs1136759 allele (G) determines a serine in the position 13 of the HLA-DRβ1 molecule located in the antigen-binding pocket. Overall, our data support the notion of unexplained SPGF as a complex trait influenced by common variation in the genome, with the SCO phenotype likely representing an immune-mediated condition.

A full list of author affiliations appears at the end of the paper.

According to recent estimations, the global prevalence of infertility has increased considerably during the last decades regardless of the socio-demographic index[1]. Specifically, up to 50 million couples worldwide currently require medical assistance for reproduction, with around half of such cases being related to male factor infertility[2,3]. Male infertility can be due either to an obstruction of the post-testicular tract or to non-obstructive causes[4]. Two extreme manifestations of the latter are non-obstructive azoospermia (NOA) and severe oligozoospermia (SO), which are characterised by a severe spermatogenic failure (SPGF) leading to a reduction in the number of spermatozoa in the ejaculate (very low concentration of spermatozoa in SO and complete lack of sperm in NOA)[2].

Many SO patients eventually father a biological child following the isolation of viable seminal spermatozoa and subsequent intracytoplasmic sperm injection (ICSI)[5]. Although this simple procedure may not be applicable to azoospermic cases, there is still a chance for men suffering from this condition to benefit from the current in vitro fertilisation techniques by undergoing a testicular sperm extraction (TESE) from a testis biopsy[6]. The overall pregnancy outcomes following TESE depend on the degree of histological abnormalities, which include hypospermatogenesis (HS, production of an extremely low number of sperm cells), maturation arrest (MA, incomplete differentiation of the germline), and Sertoli cell-only (SCO, total absence of germ cells in the seminiferous tubules). NOA patients with a histopathological diagnosis of HS have a considerably higher probability of a successful TESE when compared to those diagnosed with MA or incomplete SCO (with the latter having the poorest success rates)[7]. TESE is currently regarded as the gold standard procedure not only for sperm cell retrieval in NOA cases but also in order to obtain a conclusive histological diagnosis. However, approximately half of the TESE performed will eventually be unsuccessful in retrieving viable spermatozoa for ICSI. To that extent, having a non-invasive diagnostic test which could be able to predict sperm retrieval outcomes would be beneficial for the clinical management of NOA cases[6].

Known genetic causes of SPGF include karyotype anomalies (e.g. Klinefelter syndrome), microdeletions of the azoospermia factor (AZF [MIM 415000]) regions in the Y-chromosome, and point mutations in master regulator genes for spermatogenesis[4]. However, thus far, a genetic cause can only be established in about 20% of infertile men due to SPGF, being the origin of the infertility of the remaining cases defined as unexplained (idiopathic)[8]. In this regard, increasing evidence clearly suggests that common variants in the genome, such as single-nucleotide polymorphisms (SNP), may play a relevant role in the development of this form of male infertility by unbalancing the molecular network that controls the spermatogenic process[2,9].

Over the past decade, genome-wide association studies (GWASs), in which hundreds of thousands to millions of genetic colourblindnessvariants across the genome are interrogated in a hypothesis-free fashion, have allowed to gain a valuable knowledge about the genetic component of many complex diseases and traits[10,11]. Nevertheless, the field of SPGF research has yet to have benefited to its fullest potential from the fast progress achieved during the golden era of GWASs, likely due to the fact that most efforts have been dedicated to identifying high-penetrance rare mutations through targeted sequencing methods[9]. In this context, only three GWASs of SPGF have been performed to date, i.e. a pilot study in a population of European descent in 2009 and two well-powered studies in Asians in 2011 and 2012[12–14]. The first study did not yield consistent results due to the lack of statistical power to detect signals with a robust effect, as only 92 infertile men due to SPGF (including 52 SO and 40 NOA patients) and 80 normozoospermic controls were analysed for 370,000 SNPs[12].

Conversely, the other two Asian GWASs of SPGF, together with an additional follow-up study from one of the research groups (in which thousands of individuals were included)[15], identified several risk variants for NOA susceptibility at the genome-wide level of significance. The SPGF-associated *loci* known to date at this significance threshold map within eight genomic regions encompassing protein arginine methyltransferase 6 (*PRMT6* [MIM 608274]), peroxisome biogenesis factor 10 (*PEX10* [MIM 602859]), SRY-box 5 (*SOX5* [MIM 604975]), major histocompatibility complex, class II, DR-alpha (*HLA-DRA* [MIM 142860]), butyrophilin-like protein 2 (*BTNL2* [MIM 606000]), CDC42-binding protein kinase, alpha (*CDC42BPA* [MIM 603412]), interleukin 17A (*IL17A* [MIM 603149]), and actin-binding LIM protein family, member 1 (*ABLIM1* [MIM 602330])[13–15]. However, most of these genetic associations with NOA have not been replicated in independent studies and the histological phenotypes are yet to be analysed[9,16].

Considering the above, we established an international collaborative effort with the aim to substantially improve the current knowledge on the genetic basis of SPGF by conducting a powerful GWAS in a large case-control cohort of European ancestry. Likewise, taking advantage of the high SNP coverage that the major histocompatibility region (MHC) has in the current genotyping arrays, we also decided to specifically interrogate this genomic region at the protein sequence level.

## Results

**Testing for association with disease susceptibility in the discovery phase.** In a first attempt to identify genetic polymorphisms that could be involved in the development of the different patterns of SPGF, we performed case-control comparisons between the different established study groups and the control population in the Iberian cohort. Association signals at the genome-wide level of significance were detected in two haplotype blocks including the SNPs rs186420734, associated with TESEneg ($P = 2.95\text{E}{-}08$, OR = 11.34, 95% CI = 4.80–26.76), and rs9271527, associated with SCO ($P = 2.41\text{E}{-}08$, OR = 2.38, 95% CI = 1.75–3.22) (Table 1 and Supplementary Fig. 1). According to Open Targets, the genes functionally implicated by these variants were follicle-stimulating hormone receptor (*FSHR* [MIM 136435]) for rs186420734 and several MHC class II genes, including *HLA-DRB1* (MIM 142857) and *HLA-DRA*, for rs9271527.

Considering the strong genetic association observed between the MHC system and the SCO phenotype in our discovery cohort, we decided to conduct a more comprehensive analysis of this genomic region by inferring multiallelic SNPs, classical HLA alleles, and polymorphic amino acid positions (Supplementary Data 1). The top SCO-associated peak was observed in the MHC class II, with the SNP rs1136759 showing the strongest signal ($P = 3.04\text{E}{-}08$, OR = 2.33, 95% CI = 1.73–3.15) (Table 1 and Supplementary Data 2). This SNP is located in the coding region of the *HLA-DRB1* gene and it determines a serine in position 13 of the encoded protein (which also showed the same effect and statistical significance in the analysis), which lies in the antigen-binding pocket (Supplementary Data 2 and Fig. 1). This amino acid defines the *HLA-DRB1 13* haplotype, which represented the most associated MHC classical allele with SCO in our study cohort ($P = 3.86\text{E}{-}05$, OR = 2.19, 95% CI = 1.51–3.17) (Supplementary Data 2). No additional associations with any of the SPGF patterns analysed were observed at the genome-wide significance level (Supplementary Fig. 1).

**Replication phase in an independent population.** In order to evaluate the consistency of our results in Iberians in an

**Table 1 Genetic variants associated with spermatogenic failure subtypes at the genome-wide significance level ($P < 5E-08$) in the Iberian discovery cohort and/or in the meta-analysis by the inverse variance method with the German replication cohort.**

| Variant ID | Position (GRCh38) | Locus | Associated group | Ref allele | Iberian | | | Germany | | | Meta-analysis | |
|---|---|---|---|---|---|---|---|---|---|---|---|---|
| | | | | | Allele freq (cases/controls) | P | OR [95% CI] | Allele freq (cases/controls) | P | OR [95% CI] | P | OR [95% CI] |
| rs186420734 | 2:49429854 | FSHR | TESEneg | A | 4.96/0.54 | 2.95E−08 | 11.34 [4.80–26.76] | 0.81/0.79 | 9.82E−01 | 1.02 [0.22–4.60] | 1.37E−06 | 6.29 [2.98–13.27] |
| rs1136759 | 6:32584354 | HLA-DRB1 | SCO | G | 59.62/39.63 | 3.04E−08 | 2.33 [1.73–3.15] | 43.64/35.66 | 2.38E−02 | 1.39 [1.05–1.86] | 4.62E−08[a] | 1.78 [1.45–2.19] |
| rs115054029 | 14:97135961 | VRK1 | SCO | T | 11.54/3.80 | 6.53E−06 | 3.05 [1.88–4.96] | 4.55/1.52 | 1.79E−03 | 3.38 [1.57–7.26] | 4.24E−08 | 3.14 [2.09–4.74] |

The results of the two independent studies are also shown.

OR per-allele odds ratio for the reference allele, CI confidence interval.

[a]Results of the combined logistic regression analysis adjusted by the 10 first principal components and the country of origin: $P = 1.32E-08$, OR = 1.80, 95% CI = 1.47−2.21.

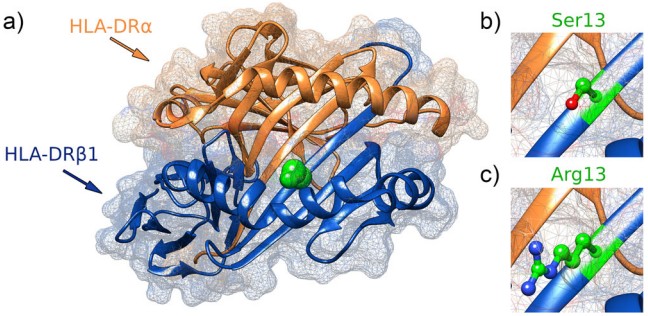

**Fig. 1 Ribbon representation of the MHC class II molecule HLA-DR.** The position 13 of the HLA-DRβ1 subunit is highlighted in green (**a**). A magnified molecular representation of the residues conferring risk (serine) and protection (arginine) to Sertoli cell-only phenotype is also shown (**b, c**).

independent European population, we generated genome-wide genotyping data in a case-control cohort from Germany. This new analysis yielded no significant genetic association of the *FSHR* region with TESEneg (rs186420734: $P = 0.98$, OR = 1.02, 95% CI = 0.22–4.60) (Table 1). Consequently, the significant *P*-value observed in the TESEneg vs. controls comparison in the Iberian population was lost in the meta-analysis including both studies (rs186420734: $P_{META} = 1.37E-06$, OR = 6.29, 95% CI = 2.98–13.27) (Table 1 and Supplementary Fig. 2), which showed a high heterogeneity between the ORs ($Q = 6.5E-03$). However, it should be noted that the lowest *P*-value across this genomic region in the German dataset was observed for rs28410762 ($P = 2.79E-04$, OR = 0.34, 95% CI = 0.19–0.61), which is located nearby the association peak in Iberians (49,399,835 and 49,429,854 in chromosome 2 for rs186420734 and rs28410762, respectively) and it is in LD with it, according to the 1KGPh3 EUR data ($D' = 1.00$, $r^2 = 0.0027$). On the other hand, a second suggestive peak of association with TESE outcome inside the *FSHR* gene was observed separately in each study as well as in the meta-analysis (top signal: rs77472631, $P_{META} = 2.95E-05$, OR = 3.18, 95% CI = 1.85–5.47) (Supplementary Fig. 2). In this case, the effect size was homogenous between populations ($Q = 0.96$).

On the contrary, the SCO-specific association signal with the MHC class II region observed in the Iberian population was replicated in the German dataset at the nominal level for this phase (rs1136759/HLA-DRβ1 Ser13: $P = 2.38E-02$, OR = 1.39, 95% CI = 1.05–1.86) (Table 1 and Supplementary Data 2). Although some heterogeneity in the ORs was observed between studies ($Q = 0.015$), consistent OR directions (towards risk) of the minor allele (G)/associated residue (Ser) were observed in both populations ($OR_{IBERIANS} = 2.33$, $OR_{GERMANS} = 1.39$). Therefore, the meta-analysis by the means of the inverse variance method confirmed this association at the genome-wide level of significance ($P_{META\ [INV\ VAR]} = 4.62E-08$, OR = 1.78, 95% CI = 1.45–2.19) (Table 1). The lowest *P*-value in the meta-analysis amongst the classical MHC alleles was also observed for *HLA-DRB1 13* ($P = 8.07E-07$, OR = 1.96, 95% CI = 1.50–2.56) (Supplementary Data 2).

In order to carry out dependency analyses in the combined population, we decided to conduct another meta-analysis using logistic regression analysis assuming an additive model adjusted by the 10 first PCs and the country of origin. A slightly more significant association between SCO and rs1136759/HLA-DRβ1 Ser13 was observed with this method ($P_{META\ [LOG\ REG]} = 1.32E-08$, OR = 1.80, 95% CI = 1.47–2.21) (Supplementary Data 2). As observed in the discovery phase, conditioning by the top signal also decreased substantially the statistical significance of class II suggestive signals (Fig. 2, Supplementary Data 2).

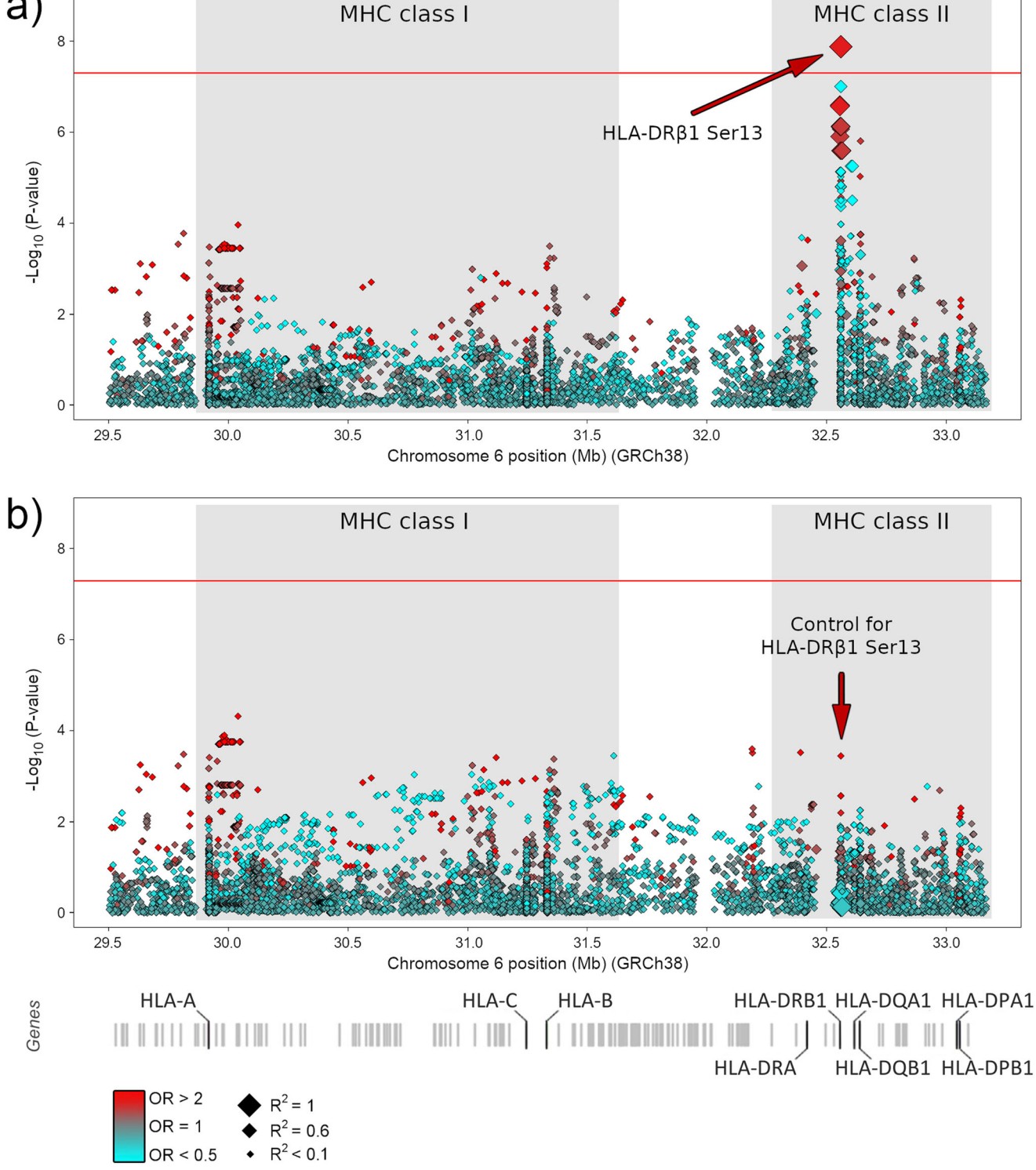

**Fig. 2 Manhattan plot representation of the logistic regression test of the MHC region accordingly with Sertoli cell-only phenotype. a** Unconditioned test of the MHC region. **b** Results of the MHC region after conditioning on HLA-DRβ1 Ser13. The −log10 of the combined logistic regression test *P*-values are plotted against their physical chromosomal position. A red/blue colour gradient was used to represent the effect size of each analysed variant (red for risk and blue for protection). The red line represents the genome-wide level of significance ($P < E{-}08$).

Similarly, when we tested the possible influence of the polymorphic amino acid positions in SCO predisposition in the combined dataset by the means of a likelihood-ratio test, the most associated position was HLA-DRβ1 13 ($P = 2.90E{-}07$). The effect sizes of the six possible residues that can be present at this amino acid position are shown in Table 2. Consistent with the above, the statistical significance of most positions was considerably reduced when conditioning on HLA-DRβ1 13, which supported the causality of this amino acid position (Supplementary Data 3 and Supplementary Fig. 3).

**Table 2 Effect on the susceptibility to Sertoli cell-only (SCO) phenotype of the residues at position 13 of the HLA-DRβ1 molecule.**

| Residue | Frequency (%) | | P | OR [95% CI] | Classical MHC alleles |
|---|---|---|---|---|---|
| | SCO | Controls | | | |
| Ser | 51.40 | 37.75 | 1.32E−08 | 1.80 [1.47-2.21] | DRB1*03, DRB1*11, DRB1*13, DRB1*14, |
| Arg | 8.65 | 14.07 | 7.93E−04 | 0.55 [0.38-0.78] | DRB1*15, DRB1*16 |
| Tyr | 11.45 | 15.43 | 3.86E−02 | 0.72 [0.52-0.98] | DRB1*07 |
| Gly | 3.51 | 5.36 | 9.71E−02 | 0.63 [0.37-1.09] | DRB1*08, DRB1*12 |
| Phe | 11.21 | 13.40 | 1.96E−01 | 0.81 [0.59-1.11] | DRB1*01, DRB1*09, DRB1*10 |
| His | 13.79 | 13.99 | 7.26E−01 | 0.95 [0.71-1.27] | DRB1*04 |

The results of the combined analysis by logistic regression and the classical MHC alleles in our dataset that contain those amino acids are shown.
OR odds ratio for the reference residue, CI confidence interval.

**Genome-wide meta-analysis of discovery and replication studies.** Taking advantage of the availability of GWAS data for the replication cohort, we aimed to identify possible additional association signals by performing a much more powerful genome-wide combined analysis using the inverse variance method (Supplementary Fig. 1).

A new genetic association at the genome-wide level of significance was observed between the SCO phenotype and a group of SNPs in complete LD with rs115054029 ($P_{\text{META [INV VAR]}} = 4.24\text{E}−08$, OR = 3.14, 95% CI = 2.09–4.74) (Table 1 and Fig. 3). In this case, the ORs were consistent between studies, with no significant heterogeneity observed ($OR_{\text{IBERIANS}} = 3.05$, $OR_{\text{GERMANS}} = 3.38$, Q = 0.82) (Table 1). The nearest gene to this haplotype is vaccinia-related kinase 1 (VRK1, MIM [602168]), which encodes a member of the VRK family of serine/threonine protein kinases playing a crucial role in regulating the cell cycle[17].

Although several suggestive signals were observed, the analyses of the remaining SPGF groups did not produce any additional significant results (Supplementary Fig. 1).

**Additional evidence of the consistency of the Sertoli cell-only phenotype associations.** The exclusion criteria for participating in this study considered known causes of male infertility that can be assessed during the clinical routine. Regarding the congenital causes, those include karyotype analysis and screening for Y chromosome microdeletions. However, the presence of high-penetrance point mutations in key genes for spermatogenesis is not usually evaluated. As a consequence, it is likely that our study cohort contained some patients of SPGF whose aetiology could be explained by a single-gene mutation. Therefore, in order to evaluate the consistency of the observed SCO genetic associations, we decided to repeat the SCO analysis after removing cases with potential monogenic causes of their infertility. With that aim, we followed a validated workflow[18] to detect the presence of rare coding variants in genes with known mutations associated with SCO, accordingly with both the "Male Infertility Genomic Consortium (IMIGC) database" and the "Infertility Disease Database (IDDB)"[19,20]. This method allowed us to identify 32 carriers of rare variants located in the exons of 40 SCO-associated genes. Interestingly, despite the evident reduction in the statistical power of this new genetic association test, the analysis of our GWAS data after removing such individuals produced even more significant results for both the HLA region (rs1136759: $P = 1.04\text{E}−08$, OR = 1.90, 95% CI = 1.52–2.36) and the VRK1 locus ($P = 3.91\text{E}−08$, OR = 3.36, 95% CI = 2.18–5.18).

**Overlap of functional annotations with VRK1 variants.** According to the variant-to-Gene (V2G) pipeline of Open Targets (which considers evidence of functionality such as QTL experiments, chromatin interaction experiments, in silico functional predictions, and distance between the variant and each gene's canonical transcription start site), all the SCO-associated SNPs in chromosome 14 were annotated as being functionally implicated in VRK1. To characterise the possible functional impact of this genomic region on SCO susceptibility, we identified all variants in high LD ($r^2 > 0.8$) with the rs115054029 haplotype in the European population of the 1KGPh3 project, considering all proxies equally as candidates for exerting the pathogenic effect, as in the previous studies[21].

Interestingly, overlap with different regulatory marks was observed for most proxies in multiple tissues (Supplementary Data 4). It should be noted that according to the ENCODE testis assays ENCFF651APG and ENCFF300WML, the proxies rs148465384 and rs17770386 ($r^2 = 1$ and 0.97 with the lead SNP rs115054029, respectively) overlap with a protein binding site for the polymerase II, RNA, subunit A (POLR2A [MIM 180660]), and rs78543559 ($r^2 = 1$ with the lead SNP rs115054029) is located in a CCCTC-binding factor (CTCF [MIM 604167]) site in the adult testis. Out of these three SNPs, rs17770386 showed a CADD value = 11.61, which predicts a high probability of deleteriousness.

In addition, accordingly, with position weight matrix (PWM) data generated from ENCODE transcription factor binding experiments[22], rs76150492 ($r^2 = 1$ with the lead rs115054029) was predicted to modify the binding site of the protein encoded by paired box gene 5 (PAX5 [MIM 167414]), which is reported to play a relevant role in spermatogenesis[23].

The possible effect of the rs115054029 haplotypic block on the deregulation of VRK1 function is consistent with the expression data of this gene reported in the Human Protein Atlas portal[24,25], which includes data from GTEx and Single Cell Expression Atlas projects[26,27], amongst others. In this regard, this gene shows an abundant expression in the testis tissue, specifically within the seminiferous ducts (Supplementary Fig. 4). At the cellular level, spermatogonia and spermatocytes show the most enhanced mRNA expression of VRK1 amongst all cell types analysed (Supplementary Fig. 4), thus suggesting a possible role of its encoded protein in the first stages of the spermatogenic process.

**Functional annotation enrichment analysis of the grey zones.** Functional annotation enrichment analysis is a powerful strategy to identify relevant cell types or tissues for a particular trait. Therefore, we evaluated the possible enrichment of DHS hotspots within the grey zone of the GWAS results (defined as the signals with P-values ranging from 1E−05 to 5E−08) for SPGF and the different histological subsets/TESE outcomes. No statistically significant enrichment was observed for any of the analysed subgroups either in the Iberian or German cohorts separately.

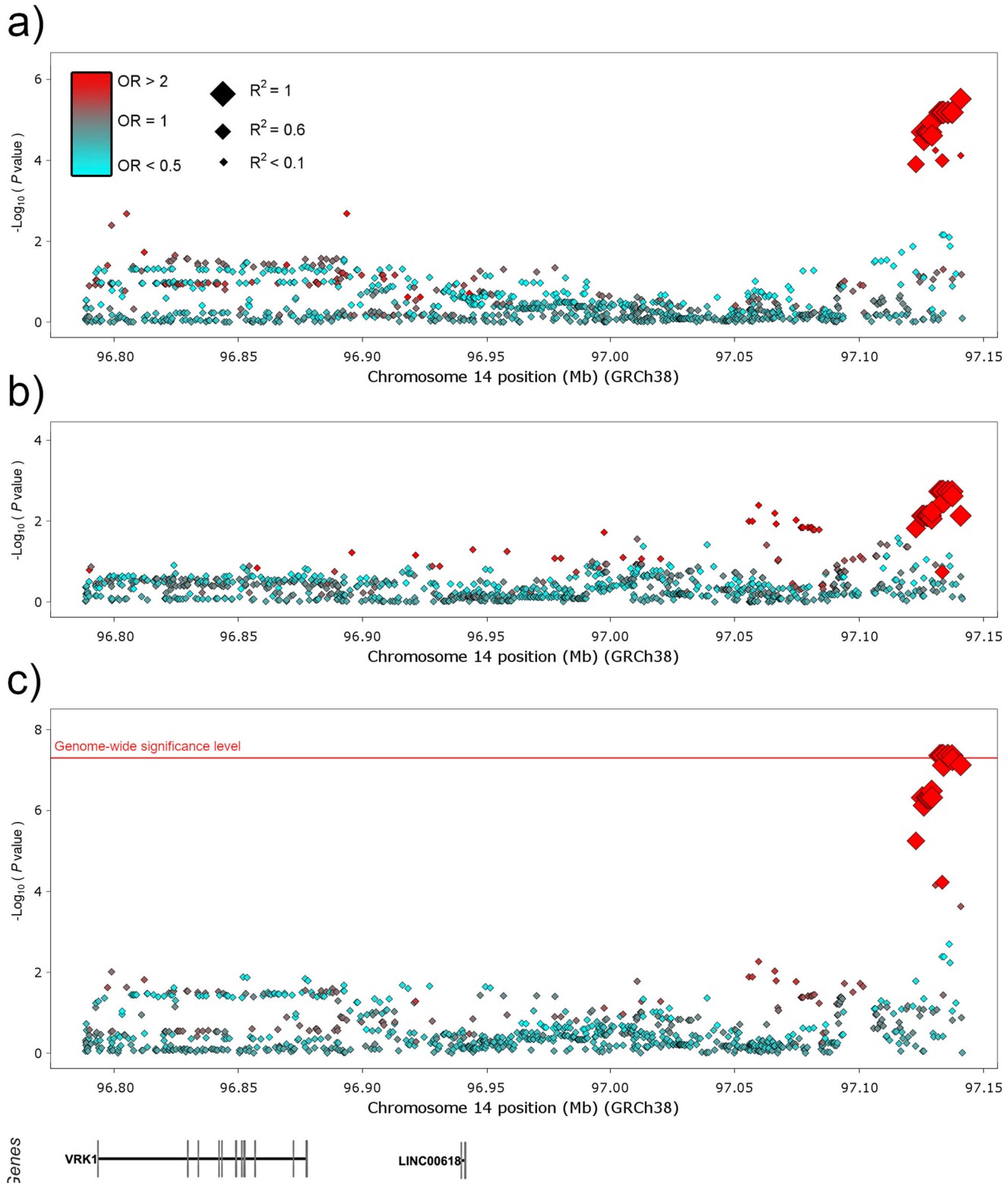

**Fig. 3 Manhattan plot representation of the logistic regression test for the *VKR1* region accordingly with Sertoli cell-only phenotype.** Data for the Iberian discovery cohort (**a**), the German replication cohort (**b**), and the combined cohort (**c**) are shown. The −log10 of the *P*-values from the logistic regression tests and the inverse variance method are plotted against their physical chromosomal position. A red/blue colour gradient was used to represent the effect size of each analysed variant (red for risk and blue for protection). The red line represents the genome-wide level of significance ($P < 5E-08$).

However, the analysis of the summary stats for the meta-analysis showed a significant DHS hotspot enrichment in SCO. Strikingly, such enrichment was specific for blood-related samples, namely CD19+ primary cells, CD20+ cells, foetal spleen, CD19+ primary cells, and GM06990 lymphoblastoid cell line (Fig. 4). The DHS enrichments detected in the analysis of the remaining combined subgroups did not reach the statistical significance (Supplementary Figs. 5–7).

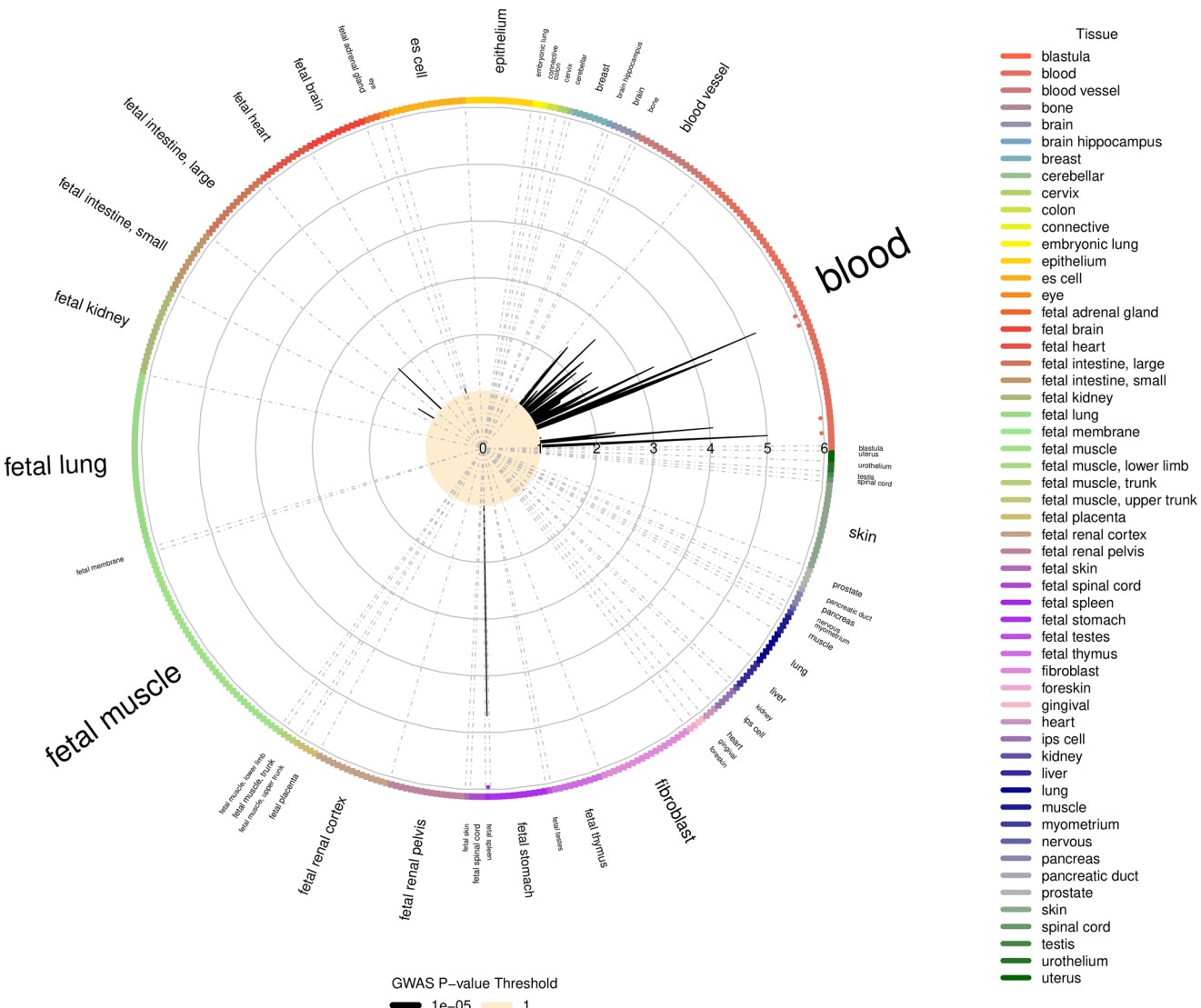

**Fig. 4 GARFIELD functional enrichment analysis of the GWAS results accordingly with Sertoli cell-only phenotype.** The radial axis represents the enrichment (OR) for each of the analysed cell types that are sorted by tissue along the outside edge of the plot. Boxes forming the edge are coloured by tissue. Enrichment is calculated for the GWAS *P*-value threshold *P* < 1E−05. Dots in the inner ring of the outer circle denote significant GARFIELD enrichment after multiple-testing correction for the number of effective annotations and are coloured with respect to the tissue cell type tested (font size of tissue labels reflects the number of cell types from that tissue).

**Previously reported associations with non-obstructive azoospermia in our dataset.** Finally, we checked in our dataset the statistical significance of non-MHC *loci* that have been described to be associated with NOA at the genome-wide level of significance (±0.5 Mbp 3' and 5' of the reported SNP) in previous studies performed in populations of Asian descent[9]. The effect size and *P*-value of both the reported association signals and the top signals observed in our combined GWAS accordingly with NOA and the extreme phenotype SCO for each region are summarised in Supplementary Data 5 and 6, respectively. Regional Manhattan plots of each genomic region are also available in Supplementary Figs. 8–10. Amongst the six analysed SNPs, only the rs13206743 variant, located in the *IL17A* genomic region at chromosome 6, showed evidence of association with NOA at the 5% significance level under the additive model (*P* = 2.32E−03), with an effect of the minor allele similar to that reported in the original Chinese study (OR = 1.20 in the present GWAS vs. OR = 1.35 in the study by Hu et al.[15]). However, suggestive *P*-values were detected across most genomic regions (Supplementary Data 6 and Supplementary Figs. 8–10).

## Discussion

We performed a genome-wide screening of around 7 million common variants in a large European cohort of well-characterised infertile men, comprising a total of 1274 patients diagnosed with SPGF of unexplained origin (772 NOA and 502 SO) and 1951 unaffected controls. The only available GWAS of this condition on this ancestry was published in 2009, which included a modest number of genetic variants and a small study cohort[12]. Therefore, we consider that our study provides an important contribution to the current knowledge on the genetic basis of SPGF, since the European population used in the previous study was underpowered, and the data on Asian populations were not analysed according to specific phenotypic patterns[12–15].

We were able to identify *VRK1* as a potential susceptibility *locus* for SCO, which represents the most severe manifestation of SPGF[28]. However, it is important to note that this association was not detected in the discovery phase but in the meta-analysis of both study cohorts. Consequently, additional replication studies

in independent populations are definitively needed before establishing *VRK1* as a firm SCO gene. *VRK1* encodes a serine/threonine protein kinase that plays a pivotal role in the regulation of the cell cycle by phosphorylating relevant transcription factors for cell proliferation such as the tumour protein p53 (MIM 191170), histones, and different proteins involved in DNA damage response pathways[29–33]. Indeed, overexpression of *VRK1* has been observed in many types of tumours, as it is directly implicated in the entry of the G1 phase of the cell cycle, chromatin condensation, Golgi fragmentation, and assembly of the nuclear envelope[34].

The human testis represents the structure with the highest expression of *VRK1* amongst all the tissues analysed in the GTEx project[27]. At the single-cell level, *VRK1* expression has been restricted to actively dividing cells of the testis (mainly spermatogonia and primary spermatocytes)[35]. In this context, the association between SCO and the *VRK1* region described here is consistent with previous studies on animal models of *VRK1* defficiency, including *Caenorhabditis elegans*, *Drosophila melanogaster*, and *Mus musculus*, all three characterised by mitotic defects with resultant infertility[36–38]. Regarding the latter, mice containing hypomorphic alleles of this gene showed reduced testis size with a progressive loss of cellularity within the seminiferous tubules and absence of spermatogenesis with increasing postnatal age. Interestingly, by 11 weeks of age, these *Vrk1*-deficient mice developed an SCO-like phenotype, with the tubules comprising only one basal layer of Sertoli cells[36]. Therefore, it is likely that the SCO-associated genotypes in the upstream vicinity of the *VRK1 locus* identified in our study cohort increase SPGF risk by altering the correct regulation of this gene. Functional experiments focused on this genomic region may shed more light on this assumption.

On the other hand, our results reinforce the hypothesis of a crucial involvement of the MHC class II region in SPGF predisposition leading to NOA. In this sense, studies performed in Japan at the beginning of the present century reported a strong contribution of the classical MHC alleles *HLA-DRB1\*1302* and *DQB1\*0604* to NOA risk, independently from the presence of Y-chromosome microdeletions[39,40]. Later on, the two GWASs performed in Chinese populations, and the follow-up study of one of them, also highlighted this genomic region as the top associated signal with NOA across the whole genome[13–15]. Additional evidence of the major involvement of the MHC class II in NOA was also generated by two recent studies, including an independent meta-analysis and a fine-mapping of this region using GWAS data, both from Han Chinese, in which the haplotype *HLA-DRB1\*1302* was confirmed as a molecular marker for NOA[41,42].

No previous studies have specifically interrogated the MHC contribution to SPGF susceptibility under a European genetic architecture[9]. With that aim, we inferred classical MHC alleles and polymorphic amino acid positions using an imputation method that has been thoroughly validated during the last decade using different approaches[43–45]. In fact, this same imputation pipeline was recently used by Huang et al. to fine-map this genomic region using GWAS data from NOA patients of Asian descent[42]. Interestingly, our analysis in Europeans showed a significant association of the MHC region specifically with the most severe NOA phenotype (defined by SCO) instead of with NOA as a whole. The SNP variant rs1136759\*G and its encoded residue in position 13 of the HLA-DRβ1 subunit (serine), were significantly overrepresented in the SCO group compared to healthy controls in both the discovery phase and in the meta-analysis. Some heterogeneity in the effect sizes on SCO was observed between the Iberian and German populations. However, in both cases, the reference alleles (rs1136759\*G and HLA-DRβ1

Ser13) showed risk ORs, and the combined analysis by logistic regression adjusted by PCs and country of origin (and, thus, controlling for possible population effects) yielded even more significant results ($P = 1.32E{-}08$, OR = 1.80) than those obtained by the inverse variance method ($P = 4.62E{-}08$, OR = 1.78). Moreover, all of the observed effects on SCO predisposition within the MHC class II region were eliminated after conditioning either on rs1136759\*G/HLA-DRβ1 Ser13 in the independent variant test or on position HLA-DRβ1 13 in the omnibus test. All these pieces of evidence point clearly towards a firm association.

The amino acid HLA-DRβ1 Ser13 defines the *HLA-DRB1\*13* classical haplotypes, which also showed a strong genetic effect on SCO in our study. Therefore, the relevant role of the *HLA-DRB1* gene in NOA reported in Asians seems to be limited to the SCO phenotype in Europeans. A possible explanation for this observation could be that the NOA cohorts included in the Asian studies were enriched in SCO patients. However, the clinical characterisation of such populations was not included in the original publications and, therefore, we can only speculate at this point. We did not detect any significant genetic effect on SPGF within the MHC class I region, as reported in the Asian population studies[42], and a power issue could not be ruled out in this case. Moreover, since our subphenotype analyses were performed with considerably lower study cohorts, this may represent the main limitation of our study. Similarly, although most of the SPGF subtypes analysed here were correlated (that is, SPGF comprised all infertile men, NOA included all subgroups except SO, and TESEneg was composed mostly of SCO and MA individuals), we did not account for possible multiple testing effects due to the subphenotype analyses of our cohort, which may also represent a major caveat because the reported associations are close to the genome-wide significance level ($P < 5E{-}08$).

In any case, the fact that the MHC class II region reached the genome-wide statistical significance when analysing our less-powered SCO group compared to the larger NOA group gives an idea of the high impact of this region on the most extreme SPGF phenotype. In this regard, the position 13 of HLA-DRβ1 associated with SCO in our study is located in the binding groove of the HLA-DR molecule, being directly involved in the molecular interactions with the presented peptide, which implies a functional impact on T cell antigen recognition, either during early thymic development or peripheral immune responses[46]. Recent evidence also suggests that certain HLA-DRβ1 epitopes may increase the risk for autoimmune processes by favouring macrophage polarisation in an antigen-presenting-independent fashion[47]. Strikingly, this same amino acid position also represents one of the most relevant MHC positions in different immune-related diseases, including systemic lupus erythematosus, giant cell arteritis, rheumatoid arthritis, and type I diabetes, amongst others[43,48–50]. Indeed, there is firm evidence pointing to the immune response as a possible aetiological factor in SPGF. For example, (1) autoimmune responses against testicular structures and/or germ cells have been found to be associated with cryptorchidism (which may lead to SPGF), and patients of this condition carrying certain *HLA-DRB1* haplotypes have been reported to show a higher production of anti-sperm antibodies, (2) infection and inflammation of the male genital tract is frequent in men diagnosed with male infertility, (3) acute or chronic inflammation may impair the testicular function through the inhibition of steroidogenesis and disturbance of the germ cell epithelium, (4) immune cell infiltrates associated with an exacerbated immune response have been observed in testicular biopsies from NOA patients, and (5) an expression signature comprising proinflammatory genes has been correlated with NOA[51–56].

Therefore, the contribution of autoimmune processes to the extreme forms of SPGF like SCO should not be disregarded. Our data definitively support this idea and are consistent with the aetiological mechanism proposed by Gong et al., in which a chronic subclinical testicular inflammation may produce the release of novel self-peptides triggering autoimmunity through antigen-presentation to Th17 cells[57]. In fact, active chromatin regions in immune-related cell types and tissues are enriched with suggestive genetic associations with SCO (Fig. 4). Under this pathogenic scenario, it could be possible that the presence in the genome of some MHC class II genetic variants, such as rs1136759*G that implies a serine in the position 13 of HLA-DRβ1, may increase the probability of initiating such autoimmune response by favouring the presentation of more immunogenic peptides.

Finally, the association signal with unsuccessful TESE at the 5' upstream region of *FSHR* detected in our discovery cohort was not replicated in the German population. In males, the follicle-stimulating hormone (FSH) is a major regulator of testis development and spermatogenesis through binding to its receptor (FSHR), which is located in the cell membrane of the Sertoli cells[58,59]. This pathway is also very relevant in female fertility, as it controls folliculogenesis and drives oocyte maturation[60]. Consequently, increasing evidence highlights these two genes as key players in the development of infertility. Although high-penetrant inactivating mutations of this signalling pathway are scarce, several SNPs in the genes encoding FSHR and the beta subunit of the ligand (FSHB) have been associated with unfavourable reproductive parameters in both female and male subjects (including SPGF cases) in a vast number of studies[61–72]. In addition, some of those SNPs have been also reported to influence the gene expression of *FSHR/FSHB* likely by modifying transcription factor binding sites in regulatory regions[66,67,70–73]. Therefore, a combined effect of both genes in male reproductive impairment has been proposed by integrating the transcriptional activity and the receptor sensitivity, which could be affected by common variations of the *FSHB* and *FSHR* genes, respectively[74]. Consistent with the above, stratification of patients accordingly to the risk genotypes of this pathway is being considered for improving the current FSH treatments of male infertility patients, which has been shown to improve sperm parameters in SPGF men[75–77].

Taking all the above into consideration, we are confident in the consistency of the GWAS peak in the *FSHR* region detected in our analysis in Iberians accordingly with the TESE success. It can be speculated that there might be population-specific LD patterns that may link the associated rs186420734 SNP with the causal variant/s in the Iberian and German genetic backgrounds. Under this assumption, and considering that rs186420734 is a rare variant in the healthy population, a possible different tagger in Germans could not be detected due to a power limitation. This could be also the case with the seven previously reported non-MHC NOA hits at the genome-wide significance level in Asians, from which only *IL17A* rs13206743 was replicated here at the nominal level.

In conclusion, our results support the notion of unexplained SPGF as a complex trait influenced by common variation of the genome, with the added effect of risk genetic variants in an individual (mainly in non-coding regulatory regions) being critical for its development[9]. Moreover, the data presented here also suggest that SPGF (or NOA) is not a single disease from a genetic point of view, but a combination of different phenotypes that have only in common a critical failure of the spermatogenic process at different points; thus underpinning the importance of defining homogeneous study groups for elucidating its genetic basis. Therefore, there is still a long way to go until we may fully characterise the molecular network that underlies SPGF. Male infertility GWAS remain lagging behind many other fields and, indeed, much larger studies focused on specific SPGF phenotypes are still needed. An integrative approach will be also helpful in this challenging endeavour, considering the key role of the non-coding polymorphisms in SPGF predisposition and the intricate haplotype architecture of the genome. Hopefully, with time and effort, the increase in the understanding of these complex processes may help to develop more efficient diagnostic and prognostic tools that could anticipate both the diagnosis and TESE outcome before the analysis of a testis biopsy, thus preventing the NOA patients with extreme phenotypes from undergoing unnecessary surgeries.

## Methods

**Study population.** Two independent case-control cohorts of European descent (including a discovery cohort from the Iberian Peninsula and a replication cohort from Germany) were analysed in this study, comprising a total of 1274 infertile men due to SPGF of unexplained origin (772 NOA and 502 SO patients) and 1951 unaffected controls. Informed written consent was signed by all participants before being enrolled in the study and all DNA samples were irreversibly anonymised. The following procedures were in accordance with the tenets of the Declaration of Helsinki and received approval by the Ethics Committee "CEIM/CEI Provincial de Granada" (Andalusia, Spain) at the session held on January 26, 2021 (approval number: 1/21). Besides, each participating centre received ethical approval and complied with the requirements of their local regulatory authorities.

SPGF cases were recruited in different public health centres and private fertility clinics from Spain and Portugal, and at the Centre of Reproductive Medicine and Andrology, University Hospital Münster, Germany, following comprehensive selection criteria based on the approved guidelines for the management of infertile men by the American Urological Association (AUA)/American Society for Reproductive Medicine (ASRM), the Canadian Urological Association (CUA), and the World Health Organization (WHO, 2010)[78–80]. These criteria include a physical examination of male patients showing evidence of clinical infertility by revision of the medical history, genetic screening (including Y-chromosome microdeletions and karyotype analysis), endocrine profile (follicle stimulating hormone, luteinizing hormone and testosterone), and semen analysis. Patients with no signs of post-testicular ejaculatory duct obstruction were analysed to establish the diagnosis of SO (<5 million spermatozoa/mL semen) or NOA (total absence of sperm in the ejaculate after two high-speed centrifugation processes in two different semen samples).

Patients showing known causes of male infertility were excluded from the study. Consequently, as in other related genetic studies[16,21,81], only those men with a normal history of testicular development with no evidence of either testicular (such as orchitis, testicular malformations, and obstruction of vas deferens) or karyotype/chromosome abnormalities were selected. The non-obstructive primary spermatogenic impairment was subsequently confirmed in around half of our SPGF cohort by the histological analysis of a testicular biopsy from those patients that decided to undergo assisted reproduction treatments involving TESE (including both conventional TESE and micro-TESE).

The pathological anatomy results from the biopsy were used to classify the SPGF patients into different subgroups according to the observed histological phenotypes, including HS (extremely low cell counts of the germline but with all stages of spermatogenesis/spermiogenesis observable in few testicular locations), MA (early maturation arrest either at spermatogonia or at primary spermatocyte stages of more than 90% of the germline), and SCO (total absence of germ cells in all seminiferous tubules). Furthermore, two additional subgroups of NOA were established based on the TESE outcome, as follows: TESEneg (if no viable sperm cell was retrieved from the biopsy) and TESEpos (including NOA patients with a successful sperm retrieval). SO patients were not considered for this classification because the TESE success rate associated with this form of infertility is close to 100%[82]. All the available information about the main clinical features of our study cohort is shown in Supplementary Table 1.

**Generation of genotype data and quality controls.** Genomic DNA samples obtained from peripheral blood mononuclear cells of every participant were genotyped at the genome-wide level using the Infinium™ Global Screening Array-24 v3.0 (GSA) in an iScan System (Illumina, Inc), following the manufacturer's protocol. This is an advanced high-throughput genotyping platform that allows the genotyping of more than 700,000 carefully selected genetic variants, including tag polymorphisms, relevant markers for clinical research, and variants for quality control (such as ancestry informative markers). Thus, this system delivers a high genomic coverage ideal for imputation methods. The genotyping of the Iberian samples was conducted in the Human Genotyping Unit of the National Genotyping Centre (CEGEN) at the Spanish National Cancer Research Centre (Madrid, Spain), whereas that of the German samples was carried out in the Genomics Unit of the LIFE & BRAIN GmbH Biomedical & Scientific Technology Platform (Bonn,

Germany). In both cases, the genotype calling was performed with the Genotyping Module (v.2.0) implemented in the GenomeStudio software (Illumina, Inc), and assigning the chromosome positions according to the Genome Reference Consortium Human Build 38 (GRCh38).

The genotype data was subject to stringent quality control (QC) measures using R and PLINK v.1.9[83]. First, we removed all the genetic variants with a cluster separation < 0.4 and filtered out INDELs and rare variants with minor allele frequencies (MAF) < 0.01. Moreover, SNPs with call rates < 0.98 and those whose genotype distributions deviated from Hardy–Weinberg equilibrium (HWE) in controls ($P < 0.001$) were also excluded from further analyses. Regarding the QC of the recruited individuals, samples with <95% of successfully called SNPs and one subject per pair of first-degree relatives (identity by descent >0.4) were excluded. In addition, principal component (PC) analyses were conducted with a set of 2921 ancestry-informative markers included in the GSA chip, in order to detect and remove population outliers (>4 standard deviations from the cluster centroids of each population) using PLINK, R and the gcta64 software. Supplementary Fig. 11 showed the two first PCs plotted against each other for the samples that remained after the removal of population outliers.

**Imputation methods**. To maximise the genetic coverage of our data sets, we conducted SNP genotype imputation for chromosomes 1-22 and X on the genome build GRCh38, using the haplotype data of the 'NHLBI Trans-OMICs for Precision Medicine' (TOPMed) programme (freeze 5) as reference panel, in the Next-Generation Genotype Imputation Service of the TOPMed Imputation Server[84]. Eagle v.2.4. for haplotype phasing[85] and minimac4 algorithms were applied for genotype imputation[86]. Due to the lack of a Y chromosome reference panel, we could not impute additional Y chromosome variants; therefore, only the directly genotyped SNPs were analysed.

Moreover, considering previously reported evidence regarding the possible role of the major histocompatibility (MHC) system in NOA predisposition[42], we decided to carry out a more comprehensive interrogation of this genomic region in our study population. With that aim, we extracted the extended MHC region (from 29 to 34 Mbp in chromosome 6) from the non-imputed data and used the SNP2HLA method[44], with a reference panel collected by the Type 1 Diabetes Genetics Consortium comprising 5,225 individuals of European origin[87], to impute SNPs, classical MHC alleles at two- and four-digits, and polymorphic amino acid positions.

To ensure the high quality of the imputed data, only SNPs with a very reliable imputation quality metric (namely Rsq > 0.9 for minimac4 or posterior probability > 0.9 for SNP2HLA) were analysed (genotypes that did not reach the selected cut-off value were set to missing). Furthermore, the imputed data underwent also rigorous QC filters using PLINK and R, including the removal of singletons, rare variants (MAF < 0.01), and polymorphisms with call rates lower than 98%. SNPs whose genotype frequencies showed evidence of deviation from HWE ($P < 0.001$) were also excluded from further analyses.

Following the QC procedures, the final case-control data sets comprised 627 SPGF patients and 1027 unaffected controls from the Iberian Peninsula and 647 SPGF patients and 924 control participants from Germany. A total of 7,371,432 SNPs were analysed in the Iberian cohort and 7,536,533 SNPs in the German cohort. Regarding the comprehensive interrogation of the MHC region, the imputed data included 7258 SNPs, 424 classical alleles (at 2- and 4-digit coverage), and 1276 polymorphic amino acid variants from the human leucocyte antigen (HLA) genes HLA-A (MIM 142800), HLA-B (MIM 142830), HLA-C (MIM 142840), HLA-DPA1 (MIM 142880), HLA-DPB1 (MIM 142858), HLA-DQA1 (MIM 146880), HLA-DQB1 (MIM 604305), and HLA-DRB1 (MIM 142857).

**Statistics and reproducibility**. To determine the minimum effect sizes that could be detected in this study based on experimental design, power analysis estimations were calculated with the online tool of the Genetic Association Study (GAS) Power Calculator, which implements the methods described in Skol et al. assuming additive genetic effects[88] (Supplementary Table 2).

All the case-control comparisons were performed with PLINK and R. In the first step, we tested for association using the imputed data of the discovery cohort (Iberian). Specifically, we compared all case groups (SPGF, NOA, SO, MA, HS, and TESEneg) against the group of unaffected controls using logistic regression on the best-guess genotypes (Rsq > 0.9), adding the 10 first PCs and the country of origin (Spain or Portugal) as covariates and assuming additive effects. If a subtype-specific genetic association was detected, cases showing such clinical phenotype/TESE outcome were also compared against those not showing it, to check whether the association was maintained after eliminating SPGF as a possible confounding variable.

With regards to the analysis of the MHC region, we tested SNPs, classical HLA alleles, and all possible combinations of amino acid residues per position by logistic regression as described above. For the positional model analysis, we established a null generalised linear model for each position including the 10 first PCs and the country of origin as covariates, which was compared on the basis of a $\chi^2$-based estimate to an alternative model including such covariates and all the possible residues at those positions (considered as conditioning factors)[43].

Besides, considering the extensive linkage disequilibrium (LD) of this genomic region, dependency analyses were performed to identify independent genetic effects

by step-wise logistic regression with conditioning by the top association signals (together with the 10 first PCs and the country of origin).

After evaluating the relevance of the results of the discovery phase, we decided to analyse an independent replication cohort from Germany following the same workflow described above for the discovery cohort.

Finally, since whole-genome genotype data were generated for both the discovery and the replication cohorts, we decided to conduct a combined analysis of both studies by the means of the inverse variance weighted meta-analysis under a fixed effects model; thus, increasing the statistical power to detect additional association signals. In this case, the possible heterogeneity of the effect sizes between the two analysed studies was evaluated using both $I^2$ and Cochran's $Q$ tests. Additionally, we also performed a combined analysis of the MHC region (including both the discovery and the replication cohorts) by logistic regression on the best-guess genotypes (>0.9 probability) assuming an additive model with the 10 first PCs and the country of origin (Spain, Portugal, and Germany) as covariates, in order to allow an adequate evaluation of the dependency effects in the pooled dataset[43].

Odds ratios (OR) and 95% confidence intervals (CI) were calculated for all the statistical analyses. The statistical significance was set at the genome-wide level ($P < 5E−08$) in the meta-analysis, provided that the $P$-value for each study separately was below 0.05 and the directionality of effect presented by the ORs was consistent between studies. The Manhattan plots were generated using an in-house R script, and the zooms of the associated regions were created with LocusZoom.js[89]. The 3D models of the HLA molecules were performed with the UCSF Chimera software[90]. The online tools provided by the GTEx[27] and LDlink[91] portals were used for figure generation together with custom R scripts.

**In silico characterisation of the associated regions**. In order to shed light on the possible pathogenic mechanisms involved in SPGF susceptibility, we decided to enrich our results with publicly available functional annotation data by using different bioinformatics approaches.

With that aim, we first used LDLink[91] to identify all proxies of the associated variants outside the MHC region ($r^2 > 0.8$) in the European population of the 1KGPh3. Then, we queried different databases and online tools to extract all the relevant information that could help us to elucidate the potential molecular and cellular mechanisms underlying the observed associations, including RegulomeDB[92], Haploreg v.4.1.[93], Open Targets Genetics[94], SNPnexus[95], GTEx[27], Human Protein Atlas[24,25], and ENCODE[96], which integrate the datasets included in Ensembl, SIFT, Polyphen, CpG, Vista enhancers, miRbase, TarBase, TargetScan, miRNA Registry, snoRNA-LBME-DB, Roadmap, Ensembl regulatory build, CADD, DeepSEA, EIGEN, FATHMM, fitCons, FunSeq2 GWAVA, and REMM. The different predictive scores for functionality are described in Supplementary Tables 3 and 4.

Furthermore, the possible overlap of the associated variants and their proxies with regulatory regions in the testicular tissue was assessed by analysing the testis-specific assays in ENCODE[96]: DNase-seq hypersensitivity sites (ENCFF323BCL, ENCFF608KRZ); CTCF (ENCODE sample references: ENCFF300WML, ENCFF559LDF, ENCFF644JKD, ENCFF767LMP, ENCFF788RFY, ENCFF855EVV) and POLR2A (ENCFF535DHF, ENCFF651APG) protein ChIP-seqs; H3K4me3 (ENCFF286DAB, ENCFF509DBT), H3K4me1 (ENCFF316MJM), H3K27ac (ENCFF610XSK, ENCFF819NRA), H3K9me3 (ENCFF711LHL), and H3K27me3 (ENCFF881OHS) histone modification ChIP-seqs.

Finally, we also assessed the enrichment of the suggestive association signals ($P < 1E−05$) observed for the analysed phenotypes and the DNase I-hypersensitive sites (DHS hotspots) identified by ENCODE[96] and the Roadmap Epigenomics project[97] for all available cell types using GARFIELD[98]. In brief, GARFIELD performs a greedy LD-prunning, LD-based tagging, and functional annotation of the genetic variants included in the GWAS summary statistics. Functional annotation enrichment is quantified by the means of generalised linear models controlling for distance to the nearest TSS and number of LD proxies, and establishing different genome-wide significance thresholds. According to Bonferroni's multiple testing correction based on the number of independent tests carried out, the significant threshold for enrichment in DHS hotspots was established at $P$-value < $2.6E−04$, as recommended by Iotchkova et al.[98].

**Reporting summary**. Further information on research design is available in the Nature Portfolio Reporting Summary linked to this article.

## Data availability

The summary statistics of the combined GWAS analysed here (both NOA and SO) are available through the NHGRI-EBI GWAS Catalogue (https://www.ebi.ac.uk/gwas/downloads/summary-statistics). Individual-level genotype data are not publicly available because they could compromise the privacy of participants and informed consent. The 3D structure of the HLA-DR molecule shown in Fig. 1 is based on the Protein Data Bank entry 3pdo, with a direct view of the peptide-binding groove. The source data behind the plots shown in Fig. 4 and Supplementary Figs. 5–7 is included in supplementary data 7. All other data are contained either in the article file and its supplementary information or available upon reasonable request to the corresponding authors.

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

## Acknowledgements

We thank the National DNA Bank Carlos III (University of Salamanca, Spain) for supplying part of the control DNA samples from Spain and all the participants for their essential collaboration. This work was supported by the Spanish Ministry of Science through the Spanish National Plan for Scientific and Technical Research and Innovation (refs. SAF2016-78722-R and PID2020-120157RB-I00), the Andalusian Plan for Research and Innovation (PAIDI 2020) (ref. PY20_00212), and the R+D+i Projects of the FEDER Operational Programme 2020 (ref. B-CTS-584-UGR20). F.D.C. was supported by the "Ramón y Cajal" programme (ref. RYC-2014-16458), and L.B.C. was supported by the Spanish Ministry of Economy and Competitiveness through the "Juan de la Cierva Incorporación" programme (ref. IJC2018-038026-I, funded by MCIN/AEI /10.13039/501100011033), all of them including FEDER funds. A.G.J. was funded by MCIN/AEI /10.13039/501100011033 and FSE "El FSE invierte en tu futuro" (ref. FPU20/02926). IPATIMUP integrates the i3S Research Unit, which is partially supported by the Portuguese Foundation for Science and Technology (FCT), financed by the European Social Funds (COMPETE-FEDER) and National Funds (projects PEstC/SAU/LA0003/2013 and POCI-01-0145-FEDER-007274). A.M.L. is funded by the Portuguese Government through FCT (IF/01262/2014). P.I.M. is supported by the FCT post-doctoral fellowship (SFRH/BPD/120777/2016), financed from the Portuguese State Budget of the Ministry for Science, Technology and High Education and from the European Social Fund, available through the Programa Operacional do Capital Humano. ToxOmics—Centre for Toxicogenomics and Human Health, Genetics, Oncology and Human Toxicology, Nova Medical School, Lisbon, is also partially supported by FCT (Projects: UID/BIM/00009/2013 and UIDB/UIDP/00009/2020). SLarriba received support from "Instituto de Salud Carlos III" (grant DTS18/00101], co-funded by FEDER funds/European Regional

Development Fund (ERDF)—a way to build Europe), and from "Generalitat de Cata-lunya" (grant 2017SGR191). SLarriba is sponsored by the "Researchers Consolidation Programme" from the SNS-Departament de Salut Generalitat de Catalunya (Exp. CES09/020). The German cohort was recruited within the Male Reproductive Genomics (MERGE) study and supported by the German Research Foundation Clinical Research Unit 'Male Germ Cells' (DFG CRU326, grants to F.T. and J.G.). This article is related to the Ph.D. Doctoral Thesis of Miriam Cerván-Martín (grant ref. BES-2017-081222 funded by MCIN/AEI/10.13039/501100011033 and FSE "El FSE invierte en tu futuro").

## Author contributions

F.D.C. and R.J.P.-M. were involved in the conception, design, and supervision of the study. M.C.-M., F.T., A.M.L., L.B.-C., S.G.-M., A.G.-J., M.B., and R.J. participated in the methodology. M.C.-M., L.B.-C., S.G.-M., and A.G.-J. performed the formal analysis. M.C.-M., F.T., A.M.L., L.B.-C., F.D.C., and R.J.P.-M. were involved in the interpretation of the data. A.M.L., F.T., R.R.-E., N.G., S.Lu., G.R., S.S.-R., J.A.C., M.C.G., A.C., V.M., F.J.V., A.P., C.G., S.G., D.A., J.A., F.Q., C.C.-J., A.A., J.N., S.So., I.P., M.G.P., S.C., J.S.-C., O.L.-R., J.M., I.P.-C., P.I.M., F.C., A.B., J.Gr., L.B., S.Se., J.Go., S.La. and S.K. were responsible for study subject and data recruitment. M.C.-M., A.M.L., F.T., R.J.P.-M., and F.D.C. were involved in the original draft preparation. All authors revised critically and approved the final manuscript.

## Competing interests

The authors declare no competing interests.

## Additional information

Supplemental files include a pdf with eleven figures and four tables, as well as six excel data files.

[1]Departamento de Genética e Instituto de Biotecnología, Centro de Investigación Biomédica (CIBM), Universidad de Granada, Granada, Spain. [2]Instituto de Investigación Biosanitaria ibs.GRANADA, Granada, Spain. [3]Institute of Reproductive Genetics, University of Münster, Münster, Germany. [4]Instituto de Investigação e Inovação em Saúde, Universidade do Porto (I3S), Porto, Portugal. [5]Institute of Molecular Pathology and Immunology of the University of Porto (IPATIMUP), Porto, Portugal. [6]Center for Predictive and Preventive Genetics, Institute for Cell and Molecular Biology, University of Porto, Porto, Portugal. [7]Andrology Laboratory and Sperm Bank, IVI-RMA Valencia, Valencia, Spain. [8]IVI Foundation, Health Research Institute La Fe, Valencia, Spain. [9]Servicio de Urología. Hospital Universitari i Politecnic La Fe e Instituto de Investigación Sanitaria La Fe (IIS La Fe), Valencia, Spain. [10]IVI-RMA Lisbon, Lisbon, Portugal. [11]Department of Obstetrics and Gynecology, Faculty of Medicine, University of Lisbon, Lisbon, Portugal. [12]Unidad de Reproducción, UGC Obstetricia y Ginecología, HU Virgen de las Nieves, Granada, Spain. [13]CEIFER Biobanco - GAMETIA, Granada, Spain. [14]UGC de Obstetricia y Ginecología, Complejo Hospitalario de Jaén, Jaén, Spain. [15]UGC de Urología, HU Virgen de las Nieves, Granada, Spain. [16]Facultad CC Salud, Universidad Alfonso X "El Sabio", Madrid, Spain. [17]Unidade de Medicina da Reprodução, Departamento de Obstetrícia, Ginecologia e Medicina da Reprodução, Hospital de Santa Maria, Centro Hospitalar Universitário de Lisboa Norte, Lisbon, Portugal. [18]Centro de Medicina Reprodutiva, Maternidade Dr. Alfredo da Costa, Centro Hospitalar Universitário de Lisboa Central, Lisbon, Portugal. [19]Laboratory of Seminology and Embryology, Andrology Service, Fundació Puigvert, Barcelona, Spain. [20]Instituto de Parasitología y Biomedicina 'López-Neyra', IPBLN-CSIC, PTS Granada, Granada, Spain. [21]Departamento de Genética Humana, Instituto Nacional de Saúde Dr. Ricardo Jorge, Lisbon, Portugal. [22]Serviço de Genética, Departamento de Patologia, Faculdade de Medicina, Universidade do Porto, Porto, Portugal. [23]Institute of Reproductive and Regenerative Biology, Centre of Reproductive Medicine and Andrology, University of Münster, University Clinics, Münster, Germany. [24]ToxOmics - Centro de Toxicogenómica e Saúde Humana, Nova Medical School, Lisbon, Portugal. [25]Human Molecular Genetics Group, Bellvitge Biomedical Research Institute (IDIBELL), L'Hospitalet de Llobregat, Barcelona, Spain. [26]Department of Clinical and Surgical Andrology, Centre of Reproductive Medicine and Andrology, University Hospital Münster, Münster, Germany. [27]Departamento de Bioquímica y Biología Molecular I, Universidad de Granada, Granada, Spain. [28]These authors contributed equally: Miriam Cerván-Martín, Frank Tüttelmann, Alexandra M. Lopes. [29]These authors jointly supervised this work: Rogelio J. Palomino-Morales, F. David Carmona. ✉email: rpm@ugr.es; dcarmona@ugr.es

