## [Peer Review File · Communications Biology]

Reviewers' comments:

Reviewer #1 (Remarks to the Author):

In this manuscript, Cerván-Martín and colleagues conduct a genome-wide association study (GWAS) to identify genetic susceptibility to male infertility using two independent European cohorts comprising of infertile men due to unexplained spermatogenic failure (SPGF) and unaffected controls. The reviewer has statistical concerns that may affect the results and conclusion in this study. Please find specific comments below.

1. Correction for multiple testing

As described in the text including the Material and Methods section, the authors perform independent multiple GWASs for SPGF and 6 distinct disease subtypes (e.g., Sertoli cell-only (SCO) phenotype) in the discovery-phase analysis using the Iberian cohort and in the genome-wide meta-analysis using both Iberian and German cohorts. When performing these association analyses, the reviewer believe that a more stringent significance level (e.g., the multiple testing-corrected threshold) than the usual genome-wide significance level ($P < 5 \times 10^{-8}$) should be used to assess the association between SNPs and each of the 7 disease phenotypes.

2. Meta-analysis of GWASs

The authors find that SNP rs115054029 is genome-wide significantly associated with SCO phenotype in the meta-analysis of the two GWASs (Table 1). This meta-analysis has limitations including a lack of replication studies using independent cohorts and the limited sample size (214 SCO and 1951 controls). The authors should conduct additional replication analysis to confirm the association of rs115054029 with SCO using independent cohorts. If it is impossible to perform the additional replication study, the study limitations should be carefully described in the manuscript.

3. Other comment:

(1) Y-chromosomal SNPs

As described in the manuscript, Y-chromosome microdeletions is one of the known genetic causes of SPGF. In this study, the authors performed association analyses using autosomal and X-chromosomal SNPs, but the readers (including me) may be also interested in whether there is an association between Y-chromosomal SNPs and SPGF. The authors are encouraged to perform an additional association analysis using typed SNPs on the Y chromosome. Otherwise, the authors should be added the reason(s) why there is no association analysis of SPGF using Y-chromosomal SNPs into the manuscript.

Reviewer #2 (Remarks to the Author):

Review comments

Mutations in genes involved in spermatogenesis can lead to male infertility. In this manuscript, Miriam et al. show that MHC II HLA-DR β 1 (rs1136759*G) is an important mutation that causes SCO in European males, enriching the type of mutation that causes infertility in MHC II HLA-DR.

There are some comments:

1. There is a strong association between the MHC system and SCO phenotype, the author should display the phenotyping figure.
2. The authors analyzed the mutation of rs1136759*G to cause SCO, but there was no more reliable evidence. Could the authors screen out the cases with mutations and exclude the published genes that cause SCO?
3. Could the authors detect mutations at locus 13 of HLA-DR β 1 and changes in its protein structure and physical property?

REVIEWER #1:

We appreciate the appropriate comments and suggestions of the reviewer for helping us to provide a more accurate overview of our results.

1) Correction for multiple testing: As described in the text including the Material and Methods section, the authors perform independent multiple GWASs for SPGF and 6 distinct disease subtypes (e.g., Sertoli cell-only (SCO) phenotype) in the discovery-phase analysis using the Iberian cohort and in the genome-wide meta-analysis using both Iberian and German cohorts. When performing these association analyses, the reviewer believe that a more stringent significance level (e.g., the multiple testing-corrected threshold) than the usual genome-wide significance level ($P \times 10^{-8}$) should be used to assess the association between SNPs and each of the 7 disease phenotypes.

RESPONSE: We completely understand the reviewer's point of view, as our analyses were corrected for multiple testing regarding the human haplotypic architecture (that is, considering 1 million independent haplotypes) but not the subtype groups. However, we would like to kindly state the following:

- The 7 analysed groups (namely SPGF, SO, NOA, HS, MA, SCO, and TSEneg) were not fully independent (i.e., uncorrelated). For instance, SPGF comprised all individuals, NOA included all subgroups except SO, and TSEneg was composed mostly by SCO and MA individuals. Therefore, correcting for 7 additional independent tests would not be accurate.

- As described above, the commonly used genome-wide significance threshold was established based on a Bonferroni correction for the number of independent polymorphisms across the genome, accordingly with the estimated linkage disequilibrium of the human genetic variation [Dudbridge F, et al. *Estimation of significance thresholds for genomewide association scans. Genet Epidemiol. 2008;32(3):227-234*; Gao X, et al. *A multiple testing correction method for genetic association studies using correlated single nucleotide polymorphisms. Genet Epidemiol. 2008;32(4):361-369*; Pe'er I, et al. *Estimation of the multiple testing burden for genome-wide association studies of nearly all common variants. Genet Epidemiol. 2008;32(4):381-385*]. Moreover, because of the high rate of type II errors (false negative) obtained in GWASs, there are currently some authors who are proposing to use less stringent thresholds when large sample sets are analysed to reduce such false negative rate [Cheng Z, et al. *Revisiting the genome-wide significance threshold for common variant GWAS. G3 (Bethesda). 2021;11(2):jkaa056*].

- Due to the fact that $P < 5E-08$ is just an estimation, most GWASs including stratified analyses like ours did not apply further correction. Some examples are shown below:

[1] Lopez-Isac E, et al. GWAS for systemic sclerosis identifies multiple risk loci and highlights fibrotic and vasculopathy pathways. *Nat Commun. 2019;10:4955*.

[2] Lopez-Mejias R, Identification of a 3'-untranslated genetic variant of RARB associated with carotid intima-media thickness in rheumatoid arthritis: a genome-wide association study *Arthritis Rheumatol.* 2019;71(3):351–360.

[3] Chen L, et al. Genome-wide assessment of genetic risk for systemic lupus erythematosus and disease severity. *Hum Mol Genet.* 2020;29(10):1745-1756.

[4] Zhang, H. et al. Genome-wide association study identifies 32 novel breast cancer susceptibility loci from overall and subtype-specific analyses. *Nat Genet.* 2020;52(6):572-581.

[5] Nakayama A, et al. Subtype-specific gout susceptibility loci and enrichment of selection pressure on ABCG2 and ALDH2 identified by subtype genome-wide meta-analyses of clinically defined gout patients. *Ann Rheum Dis.* 2020;79:657-665.

[6] Wu Y, et al. GWAS of peptic ulcer disease implicates *Helicobacter pylori* infection, other gastrointestinal disorders and depression. *Nat Commun.* 2021;12: 1146.

- Despite the above, we agree with the reviewer that the lack of stratification-based multiple testing correction in this type of studies could raise doubts in the reader about the consistency of the results. Consequently, we have decided to acknowledge this as a limitation of our GWAS by adding the following at the end of the 6th paragraph of the discussion section:

“Similarly, although most of the SPGF subtypes analysed here were correlated (that is, SPGF comprised all infertile men, NOA included all subgroups except SO, and TESEneg was composed mostly by SCO and MA individuals), we did not account for possible multiple testing effects due to the subphenotype analyses of our cohort, which may also represent a major caveat because the reported associations are close to the genome-wide significance level ($P < 5E-08$)”.

2) Meta-analysis of GWASs: The authors find that SNP rs115054029 is genome-wide significantly associated with SCO phenotype in the meta-analysis of the two GWASs (Table 1). This meta-analysis has limitations including a lack of replication studies using independent cohorts and the limited sample size (214 SCO and 1951 controls). The authors should conduct additional replication analysis to confirm the association of rs115054029 with SCO using independent cohorts. If it is impossible to perform the additional replication study, the study limitations should be carefully described in the manuscript.

RESPONSE: The appreciation of the reviewer is completely right and, accordingly, we have changed the following sentence (2nd paragraph of the Discussion section):

“We were able to identify VRR1 as a novel susceptibility locus for SCO, which represents the most severe manifestation of SPGF”

By this one:

“We were able to identify VRK1 as a potential susceptibility locus for SCO, which represents the most severe manifestation of SPGF. However, it is important to note that this association was not detected in the discovery phase but in the meta-analysis of both study cohorts. Consequently, additional replication studies in independent populations are definitively needed before establishing VRK1 as a firm SCO gene.”

The limitation of the reduced sample size was stated in the 6th paragraph of the Discussion:

“Moreover, since our subphenotype analyses were performed with considerably lower study cohorts, this may represent the main limitation of our study.”

3) Other comment: Y-chromosomal SNPs. As described in the manuscript, Y-chromosome microdeletions is one of the known genetic causes of SPGF. In this study, the authors performed association analyses using autosomal and X-chromosomal SNPs, but the readers (including me) may be also interested in whether there is an association between Y-chromosomal SNPs and SPGF. The authors are encouraged to perform an additional association analysis using typed SNPs on the Y chromosome. Otherwise, the authors should be added the reason(s) why there is no association analysis of SPGF using Y-chromosomal SNPs into the manuscript.

RESPONSE: As the reviewer suggested, we have analysed the Y chromosome genotyped variants, which did not yield any significant results (it should be noted that carrying Y chromosome microdeletions was considered in our exclusion criteria, thus the lack of association with such mutations). The Manhattan plots of Supplementary Figure 1 include now the Y chromosome. Besides, the sentence below has been added to the M&M section (Imputation methods subsection):

“Due to the lack of a Y chromosome reference panel, we could not impute additional Y chromosome variants; therefore, only the directly genotyped SNPs were analysed.”

REVIEWER #2:

We are thankful for the input provided by the reviewer, which has definitively helped us to improve our manuscript. We hope that the reviewer will consider the modifications performed appropriate.

1) There is a strong association between the MHC system and SCO phenotype, the author should display the phenotyping figure.

RESPONSE: We are not completely sure about this suggestion, since we did not perform DNA sequencing but large-scale genotyping instead. Thus, the associated HLA polymorphisms do not represent high-penetrance monogenic mutations but susceptibility variants. Accordingly, we provided the main clinical features of the study population in Supplementary Table 7. We would like to apologise in advance if the reviewer’s suggestion was not pointing in this direction and

we may have misunderstood them. If such is the case, we would appreciate very much if the reviewer could specify further the proposed suggestion.

2) The authors analyzed the mutation of rs1136759*G to cause SCO, but there was no more reliable evidence. Could the authors screen out the cases with mutations and exclude the published genes that cause SCO?

RESPONSE: In our opinion, this is a very interesting comment and we would like to sincerely thank the reviewer for pointing it out. Our exclusion criteria considered known causes of male infertility that can be assessed during the clinical routine. Those include karyotype analysis and screening for Y chromosome microdeletions. However, high-penetrance point mutations in reported SPGF genes are not usually screened. As a consequence, it is possible that our study cohort could include patients with NOA that could be explained by a single-gene mutation that may have remained undetected. In principle, this should not be an important problem because we followed a population genetics approach, and it is expected that the presence of some non-idiopathic SPGF cases will not impact substantially the allele frequencies. In fact, if the presence of these patients added statistical noise to our analyses, it would likely cause a subtle increase in the probability of getting false negative results, but it would hardly affect the consistency of the observed associations.

Nevertheless, due to the reviewer's comment, we were encouraged to provide additional evidence to reinforce the reliability of our SCO results by identifying cases with putative monogenic causes of their infertility, in order to reanalyse our GWAS data without them. With that aim, we followed a validated workflow (Lopez-Rodrigo et al. *Reprod Biomed Online*, 2022) to detect presence of rare coding variants in genes with known mutations associated with SCO, accordingly with both the "Male Infertility Genomic Consortium (IMIGC) database" and the "Infertility Disease Database (IDDB)".

We identified 32 SCO carriers of rare variants within the exons of the 40 SCO-associated genes in IMIGC and IDDB. Remarkably, despite the reduction in the statistical power of this new test, the analysis of our GWAS after removing such individuals produced even more significant results for both the HLA region (rs1136759: $P = 1.04E-08$, OR = 1.90, 95% CI = 1.52-2.36) and *VRK1* ($P = 3.91E-08$, OR = 3.36, 95% CI = 2.18-5.18).

This information is now included in a new subsection of the Results named "Additional evidence of the consistency of the Sertoli cell-only phenotype associations".

Accordingly, three new references have been added:

18. Lopez-Rodrigo O, Bossini-Castillo L, Carmona FD, Bassas L, Larriba S. *Genome-wide compound heterozygote analysis highlights DPY19L2 alleles in a non-consanguineous Spanish family with total globozoospermia. Reproductive biomedicine online 45, 332-340 (2022).*

19. Houston BJ, et al. *A systematic review of the validated monogenic causes of human male infertility: 2020 update and a discussion of emerging gene-disease relationships. Human reproduction update 28, 15-29 (2021).*

20. Wu J, et al. IDDB: a comprehensive resource featuring genes, variants and characteristics associated with infertility. *Nucleic acids research* 49, D1218-D1224 (2021).

3) Could the authors detect mutations at locus 13 of HLA-DR β 1 and changes in its protein structure and physical property?.

RESPONSE: We understand the reviewer's interest in the potential functional impact of the SCO-associated amino acid in the position 13 of the HLA-DR β 1 molecule. However, in the case of the HLA system, the pathogenic effects are due, in principle, to modifications in the affinity for the presented antigens. It is well known that certain HLA alleles confer higher risk for specific immune-mediated conditions, while others exert protective effects, but the underlying molecular mechanisms of such associations remain obscure. Indeed, there are recent evidences that also suggest a possible influence of HLA-DR β 1 epitopes in the immune response in an antigen presentation-independent fashion [van Drongelen et al. *Scientific Reports*, 2021].

We can speculate that the presence of a serine in position 13 of the HLA-DRB1 protein may contribute to the loss of the immune tolerance in the testis either by favouring the presentation of immunogenic peptides in the context of a testicular environment or by triggering a pathogenic macrophage activation through allele-specific and antigen presentation-independent effects, which may not be the case in other tissues.

Accordingly with the above, we have included the following sentence in the 7th paragraph of the Discussion section:

"Recent evidences also suggest that certain HLA-DR β 1 epitopes may increase risk for autoimmune processes by favouring macrophage polarisation in an antigen presenting-independent fashion".

A new reference has been added to support this affirmation:

47. van Drongelen V, Scavuzzi BM, Nogueira SV, Miller FW, Sawalha AH, Holoshitz J. HLA-DRB1 allelic epitopes that associate with autoimmune disease risk or protection activate reciprocal macrophage polarization. *Scientific reports* 11, 2599 (2021).

Finally, we decided to improve the information shown in figure 1, which now includes a more detailed illustration of the atomic features of both the risk (Ser) and protective (Arg) residues for SCO.

EDITORIAL COMMENTS:

As requested by the Editor, we have tried to appropriately address all the reviewers' comments. We hope that this revised version may be suitable for publication in *Communications Biology*.

REVIEWERS' COMMENTS:

Reviewer #1 (Remarks to the Author):

This is a revised version of the manuscript by Cerván-Martín et al, and appropriately addresses my earlier concerns and suggestions in the text and the supplementary figure.

Reviewer #2 (Remarks to the Author):

This study will improve the current knowledge on the genetic basis of SPGF by conducting a powerful GWAS in a large case-control cohort of European ancestry. The author has supplemented the data and revised the article according to the comments. There is no further question or suggestion from me. It is recommended to accept.

REVIEWERS #1 and #2:

We thank both reviewers for their essential contribution to the final version of our manuscript.

EDITORIAL COMMENTS:

As requested by the Editor, we have provided a point-by-point response to all formal requirements in the attached editorial request file. We hope that this revised version may be suitable for publication in Communications Biology.